# Transcriptomic Study of Different Stages of Development in the Testis of Sheep

**DOI:** 10.3390/ani14192767

**Published:** 2024-09-25

**Authors:** Binpeng Xi, Shengguo Zhao, Rui Zhang, Zengkui Lu, Jianye Li, Xuejiao An, Yaojing Yue

**Affiliations:** 1Key Laboratory of Animal Genetics and Breeding on the Tibetan Plateau, Ministry of Agriculture and Rural Affairs, Lanzhou Institute of Husbandry and Pharmaceutical Sciences, Chinese Academy of Agricultural Sciences, Lanzhou 730050, China; w5562080w@163.com (B.X.); zhangrui@caas.cn (R.Z.); luzengkui@caas.cn (Z.L.); lijianye@caas.cn (J.L.); 2Sheep Breeding Engineering Technology Research Center of Chinese Academy of Agricultural Sciences, Lanzhou 730050, China; 3College of Animal Science and Technology, Gansu Agricultural University, Lanzhou 730070, China; zhaosg@gsau.edu.cn

**Keywords:** F1 hybrid of Southdown × Hu sheep, transcriptomic study, development of testis, spermatogenesis

## Abstract

**Simple Summary:**

The testis plays a crucial role in spermatogenesis and sperm production for the male sheep; the focus of this study was to use transcriptome to analyze the key developmental periods of an F1 hybrid of Southdown × Hu sheep, a cross between Southdown and Hu sheep. Through M0, M3 and M6 and Y1, four key-stage-two comparative screening differentially genes, and GO and KEGG enrichment analysis, we found the cAMP signaling pathway, MAPK signaling pathway, ECM–receptor interaction, PI3K-Akt signaling pathway, FOXO signaling pathway and other signaling pathways by enrichment, and south lake sheep testicular-development-related key genes have been found. These findings provide new insights into the development of sheep and help to improve sheep productivity.

**Abstract:**

Numerous genes govern male reproduction, modulating testicular development and spermatogenesis. Our study leveraged RNA-Seq to explore candidate genes and pivotal pathways influencing fecundity in an F1 hybrid of Southdown × Hu sheep testes across four developmental milestones: M0 (0 months old, newborn), M3 (3 months old, sexually immature), M6 (6 months old, sexually mature), and Y1 (1 years old, adult). Histological examination using hematoxylins and eosin staining revealed that the cross-sectional area of the spermatid tubules and the number of supportive cells increased in the other groups, as compared to the M0 group. The cross-sectional area of the vasculature and the number of supporting cells were found to be significantly increased in all other groups in comparison to the M0 group. We conducted GO and KEGG analyses of the differentially expressed genes (DEGs) in the three comparison groups and identified key pathways, including cAMP, MAPK, ECM–receptor interactions, PI3K-Akt, and FOXO signaling, which are closely related to testicular development and spermatogenesis. Notably, alternative splicing (AS) events were markedly elevated in M6 and Y1 stages. Key genes like *GATA4*, *GATA6*, *SMAD4*, *SOX9*, *YAP1*, *ITGB1* and *MAPK1* emerged as significantly enriched in these pathways, potentially orchestrating the transition from immature to mature testes in sheep. These findings offer valuable insights into male reproductive potential and can inform strategies for optimizing animal breeding.

## 1. Introduction

The primary hindrance to the advancement of the sheep industry lies in the reproductive capabilities of sheep, primarily due to the fact that most breeds experience a singular and seasonal estrus cycle, limiting their breeding potential [1]. The fertility of male sheep heavily relies on their ability to produce androgens and sperm, a process that at the genetic level is crucially regulated by protein-coding genes. These genes play a pivotal role in the critical stages of testicular development and germ cell maturation, maintaining the male sheep’s fertility [2]. Therefore, it is of great scientific significance to understand the biological process and molecular mechanism of testicular development.

Southdown sheep [3], known for precocity, fertility, adaptability, disease resistance, muscular build, high postnatal growth, and superior carcass quality, are often used as elite sires in crossbreeding [4,5]. Hu sheep [6] are renowned in China for strong fertility and adaptability, excelling as maternal parents in crossbreeding [7]. Hybridization, as a pivotal method for cultivating novel varieties, has the capacity to generate heterosis that surpasses or outperforms the characteristics of its parental strains [8]. The F1 hybrid of Southdown × Hu sheep, resulting from the crossing between Hu sheep and Southdown sheep, demonstrates superior traits in body size, daily gain, growth rate, and slaughter performance compared to their sire, the Hu sheep [9]. Fertility plays an important role in animal husbandry. Heredity is an important factor contributing to differences in reproductive rates. The meat quality of the first generation of an F1 hybrid of Southdown × Hu sheep was significantly improved, and the breeding process also underwent considerable enhancement.

As an efficient high-throughput sequencing method, high-throughput RNA sequencing (RNA-Seq) is widely used to obtain transcriptome sequences and gene expression profiles, and transcriptome information can be used to quantify gene expression patterns [10]. This technique has been used to explore reproductive mechanisms in many animals, including Tibetan Sheep [11], bovine [12] and yak [13]. Understand the metabolic activity in different development stages of the testis and transcriptome characteristics to clarify the mechanism of testicular development and spermatogenesis is crucial. Although a large number of studies have been conducted in humans and mice, there are few studies on the differential transcriptome profile of sheep testicular development, and most of them are limited to a specific period of testicular development, unable to fully reveal the genes related to testicular development, especially the differentially expressed genes (DEGs) during the newborn, sexually immature, sexually mature and adult stages. Therefore, it is necessary to research the sheep testicular transcriptome expression spectrum at key developmental stages. We performed a transcriptome analysis of testicular tissue from Southdown × Hu F1 sheep at various developmental stages (0, 3, 6, and 12 months of age) utilizing RNA-Seq and bioinformatics methodologies. The results of this analysis offer fresh insights into the regulatory mechanisms that govern testicular development and spermatogenesis in sheep.

## 2. Materials and Methods

### 2.1. Ethical Statementigures

This study has been approved by the Animal Ethics Committee (Permit No. SYXK-2019-010) of Lanzhou Institute of Animal Science and Veterinary Medicine, Chinese Academy of Agricultural Sciences (Lanzhou, China).

### 2.2. Animal Handling and Sample Collection

Twelve healthy Southdown × Hu F1 sheep were castrated in Qinghuan Mutton Sheep Breeding Company in Gansu Province. Anesthesia was induced via intramuscular diazepam (Jining Ankang Pharmaceutical Co., Ltd, Jining, China) (410 mg) and scopolamine (ChemeGen, Shanghai, China) (90.3 mg), subsequently followed by intravenous thiopental sodium (Shanghai SPH New ASIA Pharmaceutical Co., Ltd, Shanghai, China) (A10–20 mg/kg). The age of the sheep was obtained from sheep breeding records. There were 3 sheep at 0 months old (newborn, namely M0-1, M0-2, M0-3), 3 sheep at 3 months old (sexually immature, namely M3-1, M3-2, M3-3), 3 sheep at 6 months old (sexually mature, namely M6-1, M6-2, M6-3), and 3 sheep at 1 year old (adult, namely Y1-1, Y1-2, Y1-3). We removed testes from 12 sheep after anesthesia and then stored them in an RNA/DNA sample protector (Servicebio, Wuhan, China). We dissected each sheep’s testis longitudinally, collected the right testicular tissue within 3 min, froze them immediately in liquid nitrogen for 15 min, stored them at −80 °C, then extracted total RNA and protein. The other fraction was fixed in 2.5% glutaraldehyde solution for 8 h, dehydrated, transparent with xylene, and immersed in wax, before embedding and preparation for paraffin sectioning. After the study, all castrated sheep were kept in Qinghuan for breeding and fattening.

### 2.3. Preparation and Data Analysis of Testicular Tissue Sections

The testicular tissue was washed with running water, then immersed in wax blocks, cut into 5 µm sections, and stained using the hematoxylin and eosin (HE) technique. The sections were observed and photographed using an Olympus DP 71 microscope (Olympus Optical Co., Ltd., Tokyo, Japan). Image Pro-Plus 6.0 software that has been calibrated to convert pixel measurements into actual lengths was used to measure the diameter of the seminiferous tubules and the area of various cells in the transverse sections of 50 randomly selected (representation from different regions and growth stages) seminiferous tubules, and the number of various cells in 50 curved seminiferous tubules was also measured [14]. A *p* < 0.05 was considered statistically significant. 

### 2.4. Quantification and Quality of RNA

Total RNA was extracted from the testicular tissue of groups M0, M3, M6, and Y1 using TRIzol reagent (Servicebio, Wuhan, China) and RNeasy RNA purification kit (Servicebio, Wuhan, China) as instructed by the manufacturer. The Nano Drop ™ One spectro-photometer (Thermo Fisher Scientific, Waltham, MA, USA) is used to assess the purity and quantity of RNA. RNA quality was evaluated by 1% agarose gel electrophoresis. 

### 2.5. Transcriptome Sequencing

About 5 µg RNA/sample was the input material for RNA sample preparation. The indexed encoded samples were clustered using the NEB Next^®^ Ultra™ (New England Biolabs, NE, USA). Directed RNA Library preparation kit according to the protocol provided by the manufacturer. After clustering, the prepared libraries were sequenced at Illumina Nova Seq 6000 (Illumina, San Francisco, CA, USA). The sequence image data generated by the high-throughput sequencer were converted into sequence data (reads) by CASA VA base recognition to obtain FASTQ files. Then, the original RNA-Seq FASTQ data were filtered with Fastp v (v0.23.4) [15]. As a quality control software, Fastp v can quickly filter and correct FASTQ data to exclude reads containing adapters or n, and those of a low quality (quality score below 20), and produce the data after quality control. This was mapped to the sheep (Qar rambouillet v1.0) reference genome using HISAT2 [16]. The One software uses the Ferragina Manzini index to arrange DNA and RNA sequences [17].

### 2.6. Quantification of Gene Expression and Analysis

Readings for a given gene were calculated to estimate the expression of various gene transcripts. Gene expression is determined by the million per kilobase (FPKM) value [18], which is currently the most commonly used method for estimating gene expression [19]. DESeq2 software (1.20.0) was used to analyze the differential expression between the treatment group and the control group. The Benjamani–Hochberg algorithm was used to adjust the *p*-value (*p*-adj) to control the false discovery rate. |log2 (fold change) | ≥ 1 and pad j < 0.05 were significant thresholds for differential expression [20].

### 2.7. GO and KEGG Enrichment Analysis of DEGs

GO and KEGG analyses of DEGs were performed using Cluster Profiler to correct for gene length bias. KEGG is based on molecular found information database, especially by genome sequencing technology to produce a high-flux, large-scale molecular dataset, so as to understand the biological systems (http://www.genome.jp/kegg/, accessed on 21 July 2022) [21]. |log2 (fold change) | ≥ 1 and *p* < 0.05 were considered to have significantly enriched degrees in GO and KEGG terms. We selected genes for subsequent studies and used the STRING-db server (http://string.embl.de/, accessed on 12 May 2024) to analyze the differential gene networks and their differential expression trends. Cytoscape software (version 3.7.1) was used for visualization.

### 2.8. RT-qPCR Analysis for the Validation of RNA-Seq Data

Quantitative real-time-PCR (RT-qPCR) was used to detect the mRNA levels of 11 genes. The same RNA samples were used for reverse transcription using a reverse transcriptase kit (Takara, Dalian, China) to synthesize first-strand cDNA. Oligo 7.0 and Primer 5.0 were used to design the primers. Sheep β-actin (NM_001009784.3) was used as the internal control. The RT-qPCR cycling parameters were as follows: 95 °C for 3 min, followed by 40 cycles of 15 s at 95 °C, the optimized annealing temperature for 15 s, and 72 °C for 20 s; the primers are described in (Appendix A). Three biological replicates were used for each assay. The relative expression of the target genes was analyzed using the 2^−∆∆Ct^ method.

### 2.9. Statistical Analysis Quantification of Gene Expression and Analysis

All statistical analyses were conducted with SPSS 22.0 software (SPSS Inc., Chicago, IL, USA), the least significant difference (LSD with Fischer’s) method in one-way analysis of variance (ANOVA). The results were expressed as mean ± standard error. *p* value < 0.05 were considered as statistically significant.

## 3. Results

### 3.1. Histological Observation of Testis

HE staining showed stable peak indicators in adulthood. The rapid appearance of voids in seminiferous tubules at 6 months of age and their stability at Y1 is indicative of maturation in testicular development. Testicular tissue has seminiferous tubules and interstitial regions. M0 lacked sperm. The 6-month-olds and 1-year-olds had more differentiated tubules and enlarged Sertoli cells. Spermatogenic cells develop from spermatogonia to spermatids. The 1-year-olds had more spermatogenic layers and sperm near lumen (Figure 1A). Aging increased tubule diameter, cross-section, spermatogonia, and spermatocyte areas (*p* < 0.05), especially during maturity. Y1 had more spermatogonia (*p* < 0.05). Sertoli cell proliferation peaked M0-M3, stabilizing after M6 (*p* < 0.05) (Figure 1B).

### 3.2. Analysis of Gene Expression Profile during Testicular Development in Sheep

Twelve distinct cDNA libraries were constructed utilizing testicular tissue RNA, and the subsequent RNA-Seq data were summarized and statistically presented in the table below for concise analysis. A total of 72.47 G raw reads were generated from the 12 libraries, and 71.33 G of clean reads remained after quality control (Appendix A). The GC content of the 12 samples ranged from 56.12 to 72.87%, which was consistent with the base composition law. Q30 ≥ 91.45% indicated that the data were reliable and could be used for further analysis. The clean data were compared with the reference memory group, and more than 70.62% of the reads were accurately compared, with a high matching rate (Appendix A). About 5.22–12.72% of the clean reads had multiple aligned positions, and 57.90–85.7% of the clean reads had a single aligned position (Appendix A).

### 3.3. Alternative Splicing Data

In this work, using rMATS alternative splicing (AS) events can be divided into five types (http://rnaseq-mats.sourceforge.net/index.html, accessed on 21 February 2024). By identifying the type and analyzing each AS type, AS events occurred in all DEGs detected between the testicles of M0 vs. M3, M3 vs. M6, M6 vs. Y1. To determine the type of AS associated with testis genes, AS events were compared among M0 vs. M3, M3 vs. M6, and M6 vs. Y1 groups. The five known types of AS events include intron retention (RI), mutually exclusive exon (MXE), 3‘ splicing site (A3SS), 5‘splicing site (A5SS), and exon skipping (SE). All five types of AS were found in the group comparison. RI, MXE, A3SS, and SE were found. These findings indicate that SE is the most common AS event. By comparison, it was found that SE in M6 vs. Y1 group is higher than that in the other two groups, with an SE of 2961. The other identified AS events were RI (575), MXE (423), A3SS (701), and A5SS (1306), with an SE of 1936 in the M0 vs. M3 group. Other AS events are RI (384), MXE (209), A3SS (496), and A5SS (844) (Figure 2). The aforementioned discoveries underscore the pivotal role of alternative splicing (AS) in modulating the intricacies of gene expression during testicular development, particularly during the transitional phases from immaturity to sexual maturity, and subsequently from sexual to somatic maturity.

### 3.4. Analysis of Differentially Expressed Genes

A hierarchical cluster analysis of DEGs revealed a distinct clustering pattern according to their respective conditions, segregating the DEGs from the 12 libraries into four distinct clusters. It can be seen from the figure that M0 vs. M3, M3 vs. M6, M6 vs. Y1 have different expression patterns in general, but the cluster of M6 vs. Y1 obviously has similar repetitive expression commonality than the first two groups (Figure 3A). In order to clarify the difference in gene expression in testes of sheep at different developmental stages, the differentially expressed genes in four different developmental stages were compared and analyzed. | log2 (fold-change) | ≥ 1 pad j < 0.05 was used to determine DEGs. The results showed that 1954 DEGs were up-regulated and 1042 DEGs down-regulated in the M0 vs. M3 group, 431 DEGs were up-regulated and 1755 DEGs were down-regulated in the M3 vs. M6 group, and 2987 DEGs were up-regulated and 3964 DEGs were down-regulated in the M6 vs. Y1 group (Figure 3B). Among all DEGs detected, 1112 genes were only expressed in the immature group, 610 genes were only expressed in the M0 group, 33 genes were only expressed in the M3 group, 101 genes were only expressed in the M6 group, 1290 genes were only expressed in the Y1 group, and 10,925 genes were expressed in all three groups (Figure 3C).

### 3.5. GO Enrichment Analysis of DEGs 

To define the biological processes involved in M3 vs. M0, we carried out a Gene Ontology analysis of the DEGs. We found 2996 genes were assigned to 9301 GO terms, namely, 500 biological process (BP) terms, 1465 cell component (CC) terms, and 2836 molecular function (MF) terms (Figure 4(A1)). We identified the 85 most significant BP terms (*p* < 9.5 × 10^−5^); these terms included single-multicellular organism process, single-organism developmental process, regulation of metabolic process, etc. We identified the 15 most significant CC terms (*p* < 3.7 × 10^−5^); these terms included intracellular part, cell periphery, plasma membrane, etc. Lastly, we identified the four most significant MF terms (*p* < 9.4 × 10^−5^); these terms included binding, protein binding, ion binding, and protein heterodimerization activity. Similarly, we carried out a Gene Ontology analysis of DEGs in M3 vs. M6 to define the key biological processes occurring in sexually mature testes. A total of 2186 genes were assigned to 5652 GO terms, namely, 1909 BP terms, 1870 CC terms, and 1873 MF terms (Figure 4(A2)). We identified the 138 most significant BP terms (*p* < 8.1 × 10^−5^); these terms included G2/M transition of mitotic cell cycle, homologous chromosome segregation, regulation of protein modification by small protein conjugation or removal (RPM), negative regulation of gene expression, etc. We identified the 56 most significant CC terms (*p* < 7.3 × 10^−5^); these terms included intracellular non-membrane-bounded organelle, intracellular, intracellular part, intracellular organelle, nuclear part, etc. We identified the 33 most significant MF terms (*p* < 7.3 × 10^−5^); these terms included nucleoside binding, purine nucleoside binding, ribonucleoside binding, purine ribonucleoside binding, etc. To define the biological processes involved in M6 vs. Y1, we carried out a Gene Ontology analysis of the DEGs. To define the biological processes involved in M6 vs. Y1, we carried out a Gene Ontology analysis of the DEGs. We found 6951 genes were assigned to 17,088 GO terms, namely, 5740 biological process (BP) terms, 5696 cell component (CC) terms, and 5652 molecular function (MF) terms (Figure 4(A3)). We identified the 23 most significant BP terms (*p* < 8.1 × 10^−5^); these terms included cilium organization, developmental process, single-organism developmental process, spermatogenesis, anatomical structure development, etc. We identified the 56 most significant CC terms (*p* < 7.3 × 10^−5^); these terms included sperm part, cytoplasmic part, cytoskeletal part, plasma membrane part, sperm flagellum, extracellular matrix, etc. We identified the nine most significant MF terms (*p* < 9.5 × 10^−5^); these terms included nucleoside phosphate binding, nucleotide binding, hydrolase activity, cytoskeletal protein binding, and anion binding. 

The number of genes within each of the aforementioned GO categories exhibited a higher prevalence in M6 compared to Y1, indicative of an upregulation in spermatogenesis-related genes in sexually mature animals as opposed to their pubertal counterparts. The results showed that M6 was enriched in basic metabolic processes, Leydig cell differentiation, cell cycle control, cell cycle process, the G2/M transition of mitotic cell cycle, cell morphogenesis, the regulation of nuclear division, the regulation of cell cycle, reproductive process, male gamete generation, gamete generation, mitotic cell cycle, cell proliferation, cell differentiation, mitotic-related processes, androgen metabolism, and cell proliferation. Y1 was enriched in scilium movement, cilium morphogenesis, spermatogenesis, male gamete generation, nucleoside phosphate binding, cytoskeleton, the cellular component assembly involved in morphogenesis, motile cilium, sperm flagellum, and the axoneme part.

### 3.6. KEGG Enrichment Analysis of DEGs 

KEGG analysis showed that several pathways were significantly enriched. After comparing M0 with M3, 1323 pathways were enriched (Figure 4(B1)). We identified the 32 most significantly enriched pathways (*p* < 0.05); and these pathways included the FoxO signaling pathway (ko04068), ECM–receptor interaction (ko04512), Hippo signaling pathway (ko04390), MAPK signaling pathway (ko04010), PI3K-Akt signaling pathway (ko04151), etc. After comparing M6 with M3, 283 pathways were enriched (Figure 4(B2)). We identified the 13 most significantly enriched pathways (*p* < 0.05); these pathways included cell cycle (ko04110), Progesterone-mediated oocyte maturation (ko04914), oocyte meiosis (ko04114), cellular senescence (ko04218), Purine metabolism (ko00230), homologous recombination (ko03440), Influenza A (ko05164), human T-cell leukemia virus 1 infection (ko05166), etc. After comparing M6 with Y1, 2557 pathways were enriched (Figure 4(B3)). We identified the 58 most significantly enriched pathways (*p* < 0.05); these pathways included Glycolysis/Gluconeogenesis (ko00010), cAMP signaling pathway (ko04024), Focal adhesion (ko04510), Glutamatergic synapse (ko04724), Pyruvate metabolism (ko00620), HIF-1 signaling pathway (ko04066), AMPK signaling pathway (ko04152), PPAR signaling pathway (ko03320), etc.

### 3.7. Analysis of Differential Gene Networks and Their Differential Expression Trends

In this study, we related pathways for the reproduction of the genetic analysis of gene network interaction enrichment by the *SMAD4*, *YAP1*, *SOX9*, *ITGB1*, *FOXO1* found in cAMP signaling pathways, Hippo signaling pathways, the PI3K-Akt signaling pathway and *FOXO1* as the hub genes in the signal path (Figure 5A). We performed an expression trend analysis of the selected genes in five ways. An initial analysis of differential expression found that *YAP1* and *SMAD4*, *ITGB1*, *DDB1*, *GATA6*, and *GATA4* expression is consistent with the trend; apart from *RAB3B* expression, which had the highest quantity during M0, the others expressed the highest during M6 (Figure 5B).

### 3.8. qRT-PCR Validation of DEGs

To verify the RNA-Seq results, 11 DEGs were selected for confirmatory analysis using RT-qPCR (Figure 6). The results showed that there were some differences in their expression levels, but the expression patterns were consistent, indicating that the RNA-Seq data were reliable. 

## 4. Discussion

Sheep constitute a significant portion of the global livestock industry, and reproductive efficiency stands as a pivotal indicator of the profitability and income levels within this sector [22]. The testicular development and spermatogenesis of rams at various life stages are intricate processes that necessitate extensive gene transcription. At the heart of all phenotypic manifestations and biological behaviors lies the regulation of gene expression, which is primarily orchestrated through mRNA. Reproductive traits are intricately choreographed by thousands of protein-coding RNAs, intertwining to form a complex regulatory network that propels vital biological processes forward in a coordinated and orderly fashion [23]. 

Sertoli cells provide energy and nutrients for spermatogenesis and secrete a variety of substances that participate in the differentiation and maturation of spermatogenic cells to ensure the occurrence of sperm [24]. The number of Sertoli cells stabilizes in puberty. A reduction in fetal Sertoli cells significantly decreases testicular cord number, likely due to the loss of the basement membrane components they secrete [25]. In mice, Sertoli cells proliferate post-puberty arrest, achieving balance. Birth proliferation changes irreversibly alter this number, supporting the need for adequate Sertoli cells for testicular development [26]. Sertoli cells in the testis of sheep provide the necessary environment for spermatocytes. Studies have found that the diameter, area, and thickness of the epithelium of seminiferous tubules are related to the number of spermatogenic cells in seminiferous tubules. We found that the diameter and area of the convoluted seminiferous tubules and the number of Sertoli cells of the Y1 group were significantly higher than those of other groups. HE results showed that the 0-month-old male sheep only had spermatogonia, Sertoli cells and interstitial cells. Primary spermatocytes and a few sperm were found in 6-month-old rams. A large number of mature spermatids appeared in rams at 1 year of age. Yang et al. found that sperm cells and sperm were found in Hu sheep at the age of 9 months [27]. Recent studies have shown that the migration of spermatogenic cells in the seminiferous epithelium is associated with the remodeling of anchor junctions between cells during spermatogenesis, which are involved in spermatogenesis and sperm release [28]. Studies show higher germ cell count and seminiferous tubule diameter, as well as area and epithelial thickness in M6 and Y1 stages vs. others. We hypothesize a positive correlation among these factors, reflecting internal structural development and germ cell proliferation; these changes provide sufficient space and nutritional support for germ cells, thereby promoting spermatogenesis.

In this study, RNA-seq was used to build a complete dataset that explains the spatiotemporal transcriptome of testicular tissue in Southdown × Hu F1 sheep. Testicular growth and development are a key factor affecting sheep reproduction. Therefore, it is crucial to identify genes that regulate testicular growth and development. We speculate that during the developmental stage from immaturity to sexual maturity, various reproductive functions begin to emerge and sperm production begins at the same time, which is a very complex biological process that requires the joint regulation of a large number of different genes. When sheep continue to develop from sexual maturity to physiological maturity, the testis already has all the functions related to reproduction, so the developmental differences are relatively small and the number of relevant DEGs is significantly reduced. This is similar to the results of ZOU’s study on the testis of Qian bei ma goats at three different developmental stages [29]. 

In metazoans, AS plays an important role in the production of different protein products that play roles in different cellular processes, including cell growth, differentiation, and death [30]. The AS events detected in this study occurred mainly in the M0 vs. M3 and M6 vs. Y1 groups, most of which involved ES. A comprehensive analysis of AS events, as well as GO and KEGG enrichment results, can predict the effects of AS events on related gene functions. The supporting cells in the testis provide the necessary survival environment for spermatocytes; in this study, specific DNA damage-binding protein 1 (DDB1) was found to undergo alternative splicing in M0, M3, M6 and Y1 groups, indicating that the synthesis of *DDB1* is complex and plays an important role in the testicular development of sheep. Studies have found that *DDB1* deficiency in Sertoli cells leads to impaired Sertoli cell proliferation and abnormal testicular cord development [31]. Sertoli cells are key cells throughout testicular development and spermatogenesis, *DDB1* is ubiquitously expressed in almost all testicular cell types, and previous studies have shown that it is essential for germ cell development [32]. *DDB1* regulates the ubiquitination and degradation of cell cycle inhibitors such as p27 and affects the progression of the cell cycle, thereby ensuring the smooth progress of spermatogenesis [33]. We hypothesized that *DDB1* plays a key role in testicular development in sheep, specifically, a role that is essential for both testicular development and spermatogenesis.

As an organ characterized by significant transcriptional activity, the testis holds the primary function of producing male gametes through spermatogenesis, which necessitates the intricate expression and the regulation of numerous testis-enriched genes. From the results of GO analyses, important enrichment terms related to testicular development and spermatogenesis include spermatogenesis, spermatid motility, acrosome assembly, spermatid development, synaptic complexes, fertilization, and cilia- or flagellum-dependent cellular motility. Interestingly, among the above GO entries, some of those related to spermatogenesis were significantly enriched only in M3 vs. M6 and M6 vs. Y1. The number of DEGs in the testes of the Lesser Tailed Frosted Sheep was significantly higher in the testes at 2 and 6 months of age than in the testes at 6 and 12 months of age. The 2- and 6-month-old testicular DEGs were mainly associated with sexual maturation and multiple metabolic and biosynthetic pathways; DEGs in the testes at 6–12 months of age were mainly associated with metabolic and translational processes [34]. It can be seen that the sheep develops rapidly, mainly during this period, and the testis of males before sexual maturity develops into a dense structure that forms an ecological niche through the interaction of somatic and germ cells [35]. Spermatogonial differentiation and the regulation of hormonal signaling molecules ultimately promote stable spermatogenesis, providing the material basis for spermatogenesis and testicular development.

M0- vs. M3-enriched pathways included the MAPK signaling pathway, ECM-receptor interaction and the PI3K-Akt signaling pathway. In the seminiferous tubules, Sertoli cells can directly contact with spermatogenic cells at all levels and have multiple functions during spermatogenesis. Therefore, the status of Sertoli cells greatly affects spermatogenesis and testicular development [36]. These pathways are all associated with structural features involved in spermatogenesis. ECM–receptor interactions have been implicated in the development, differentiation, and maturation of male germ cells. Yin found that miR-542-3p can inhibit the proliferation of immature porcine testicular Sertoli cells and regulate the expression of downstream target genes by participating in the MAPK, cAMP and TGF-β signaling pathways, thereby regulating the growth of immature porcine testicular Sertoli cells [37]. For the M3 vs. M6 and M6 vs. Y1 groups, the cell cycle, thyroid hormone signaling pathway, cGMP-PKG signaling pathway and other pathways were enriched in all groups, and the cell cycle was related to the division and proliferation of male germ cells. Spermatogenesis cannot be achieved without the regulation of signaling pathways, and the MAPK, AMPK and cAMP signaling pathways play an important role in support cells. Both the AMPK and cAMP signaling pathways affect tight junctions and adhesion junctions, as well as supporting cell proliferation. The AMPK signaling pathway also regulates the supply of lactate. These signaling pathways combine to form a complex regulatory network for spermatogenesis. P38 MAPK regulates the proliferation, differentiation, and apoptosis of mouse spermatogenic cells, crucial for testicular tissue development, maturation, and spermatogenesis [38]. It also modulates stromal cells to influence testosterone synthesis and secretion, maintaining normal testicular structure and function [39]. The cAMP signaling pathway, through the phosphorylation of protein kinase A and subsequent activation of axonemins, plays a crucial role in enhancing sperm motility, a vital process for spermatogenesis [40]. As a potential therapeutic target, cAMP levels can be elevated by inhibiting phosphodiesterase (PDE) activity, thereby boosting testosterone synthesis and sperm motility, among other physiological processes [41]. This mechanism underscores its significance in drug development, aimed at improving male fertility. 

KEGG analysis showed that there were several common pathways between M0 vs. M3 and M3 vs. M6, and the common enriched pathways included FOXO signaling pathway, viral carcinogenesis, human T-cell leukemia virus type 1 infection, etc. Normal spermatogenesis requires the precise spatiotemporal expression and regulation of testicular Sertoli cells. To provide physiological and nutritional support for differentiated male germ cells, the quality and output of sperm depend on the number and maturation state of Sertoli cells, so the state of Sertoli cells greatly affects spermatogenesis and testicular development [42]. For M3 vs. M6 and M6 vs. Y1 groups, the cell cycle, thyroid hormone signaling pathway, cGMP-PKG signaling pathway and other pathways were enriched in all groups, and the cell cycle was related to the division and proliferation of male germ cells [43]. Testicular development is highly dependent on thyroid hormone status. Both hypothyroidism and hyperthyroidism affect testicular size and the proliferation and differentiation of Sertoli, Leyden, and germ cells, thereby affecting steroidogenesis, spermatogenesis, and male fertility [44]. Thyroid hormone levels in the early stages of testicular development can also affect testicular development and spermatogenesis by regulating the duration of Sertoli cell proliferation [45]. The FOXO protein family regulates cell cycle by enhancing inhibitors like p27kip1 and p21cip1, while suppressing cyclinD1, maintaining testicular cell balance [46]. The overexpression of Foxo1 in mouse Sertoli cells disrupts testicular development, leading to morphological defects, germ cell loss, and sperm count reduction [47]. HIF-1α reduction in varicocele rats improves seminiferous tubules, spermatogenic cell density/arrangement, reduces apoptosis, and downregulates VEGF/PI3K/Akt pathway [48]. Under hypoxia, HIF-1α upregulation activates PI3K/Akt to promote spermatogenic cell proliferation and inhibit apoptosis [49]. The PI3K-AKT pathway regulates testicular cell proliferation and apoptosis via cyclin-related *(Cyclin D1*, *CDK4/6*) and apoptosis-inhibiting proteins (Bad, Bax), crucial for cell-number balance [50,51]. It also modulates spermatocyte meiosis by influencing meiosis-related proteins (SCP3, DMC1). Hormones (FSH, androgen, estrogen) activate PI3K/AKT, impacting testicular development, spermatogenesis, and Sertoli cell proliferation, differentiation, and function via receptor binding [52]. 

In this study, we found that DEGs are significantly enriched in relevant mitochondrial regulatory pathways. Currently, cAMP signaling in the cytoplasm is thought to regulate dynamic changes in mitochondria [53]. Not only that, Follicle-Stimulating Hormone acts via the FSH-Receptor by employing cAMP as the dominant secondary messenger in testicular Sertoli cells to support spermatogenesis [54]. We also found that the *SOX 9* gene, which belongs to the Sox protein family, is enriched in this pathway [55]. In this study, it was observed that the expression of *SOX9* in the testis of sheep initially decreased and then gradually increased. The lowest expression was found at M3, while Y1 exhibited significantly higher levels compared to other stages, consistent with the findings of Li Tao Tao et al. [56]. It is speculated that the expression of SOX9 mRNA shows a steady upward trend after 6 months of age. This may be attributed to reduced external stress as the sheep age, coupled with their onset of sexual maturity and gradual improvement in reproductive function.

According to the differential gene network interaction analysis, insulin growth factor 1 (*IGF1*) and SMAD4 were closely related to other differential genes. The SMAD protein family (SMADs) is the downstream molecular target of TGFβ signaling pathway, which controls the physiological development or degradation of animal cells by regulating cell physiological activities and target gene transcription [57]. *SMAD4*, associated with testicular development and spermatogenesis, can lead to spermatogenesis dysfunction, abnormal testicular dysgenesis, and hormone secretion [58]. *YAP1* supports cell and sperm cells and stromal cells and endothelial cells expressing strong Yang, and *YAP1* inactivation results in the decrease in Sertoli cells in male mice gene expression [59]. These observations are consistent with previous research, namely the mitochondrial function and Hippo pathway in control of cell proliferation and organization steady-state crosstalk [60]. We hypothesized that components of the Hippo pathway might serve as a new pathway to regulate the male reproductive development potential of sheep.

## 5. Conclusions

In this study, RNA sequencing was used to analyze the transcriptome and histome of sheep at four developmental stages (M0, M3, M6 and Y1). Significant differences in testicular histome were found between puberty and adulthood, and specific genes involved in testicular development and spermatogenesis were identified among the three comparison groups. Genes such as *FOXO1*, *SMAD4*, *GATA4*, and *YAP1* play important roles in physiological processes such as testicular development and spermatogenesis through terms such as cAMP, PI3K-Akt signaling pathway, FOXO signaling pathway, HIF-1 signaling pathway, etc. For example, these genes can be defined as candidate genes controlling testicular development and spermatogenesis. The current study provides an overview of detailed transcriptomic changes in the ovine testis at different developmental stages and provides new insights into the differentially expressed genes associated with spermatogenesis this study.

## Figures and Tables

**Figure 1 animals-14-02767-f001:**
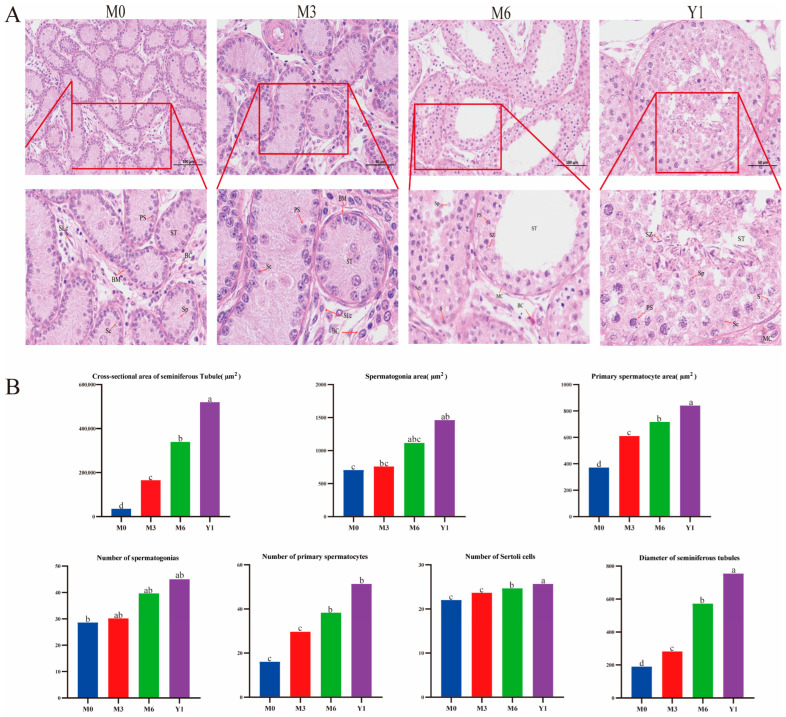
(**A**) Histological observation of testicular tissue of sheep at different stages of development. ST: seminiferous tubules; BC: capillaries; BM: basement membrane; MC: myoid cells; S: spermatogonia; PS: primary spermatocytes; SZ: sperm; SP: spermatids; Sc: sertoli cells. (**B**) Comparison of the histological parameters of testicular seminiferous tubules of sheep at different stages of development. Different letters in the same row indicate significant differences (*p* < 0.05).

**Figure 2 animals-14-02767-f002:**
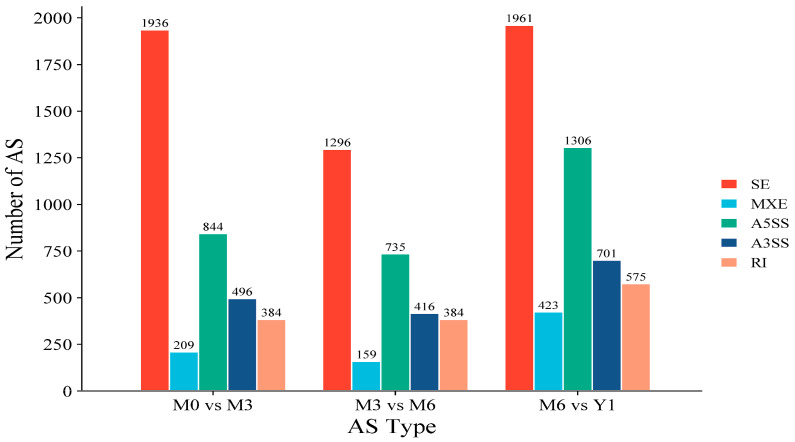
Comparison of the histological parameters of testicular seminiferous tubules of sheep at different stages of development.

**Figure 3 animals-14-02767-f003:**
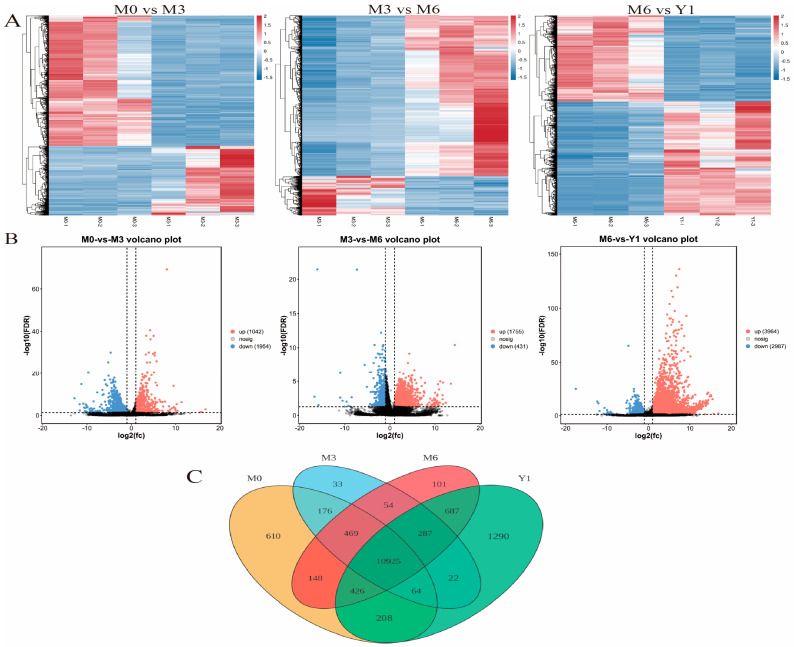
(**A**) Clustering of differentially expressed genes. log10 (FPKM + 1) was used to normalize the clustering of the FPKM system. (**B**) Volcano map of differentially expressed genes (DEGs). The red, blue, and black dots in the figure represent significantly up-regulated, down-regulated, and unchanged transcripts, respectively. (**C**) Venn diagram depicting gene expression patterns. It is given that (FPKM > 1) The number of unique and co-expressed genes. M0 represents 0-month-old, M3 represents 3-month-old, M6 represents 6-month-old, Y1 represents 1-year-old.

**Figure 4 animals-14-02767-f004:**
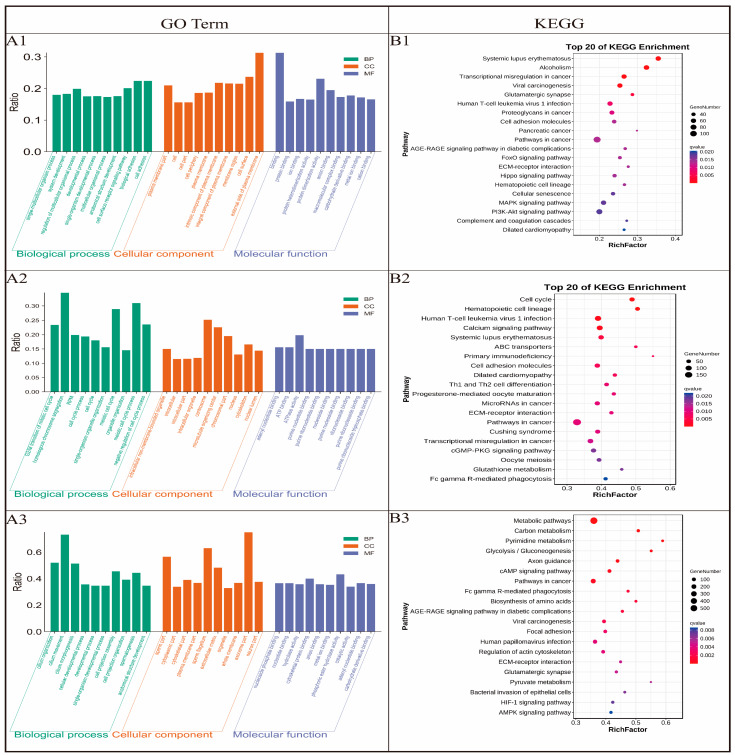
(**A**): GO annotation of the differentially expressed genes. (**B**): KEGG enrichment analysis of the differentially expressed genes. (**A1**,**B1**): M0 vs. M3, (**A2**,**B2**): M3 vs. M6, (**A3**,**B3**): M6 vs. Y1.

**Figure 5 animals-14-02767-f005:**
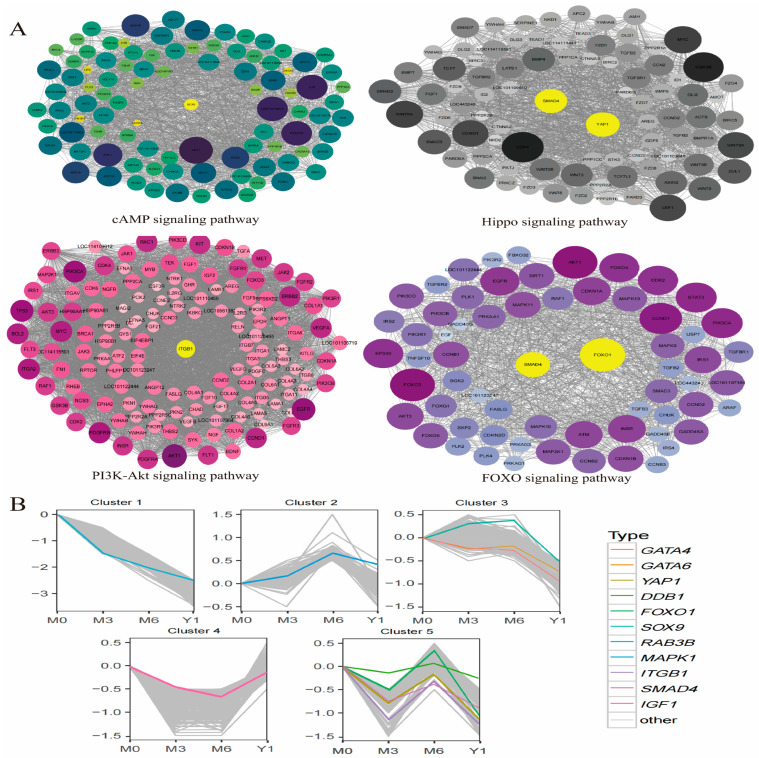
(**A**) Protein–protein interactions (PPIs) among genes enriched in reproduction-related pathways, where yellow circles signify high-degree nodes and the remaining circles represent low-degree nodes. (**B**) A trend chart illustrating differences in gene expression.

**Figure 6 animals-14-02767-f006:**
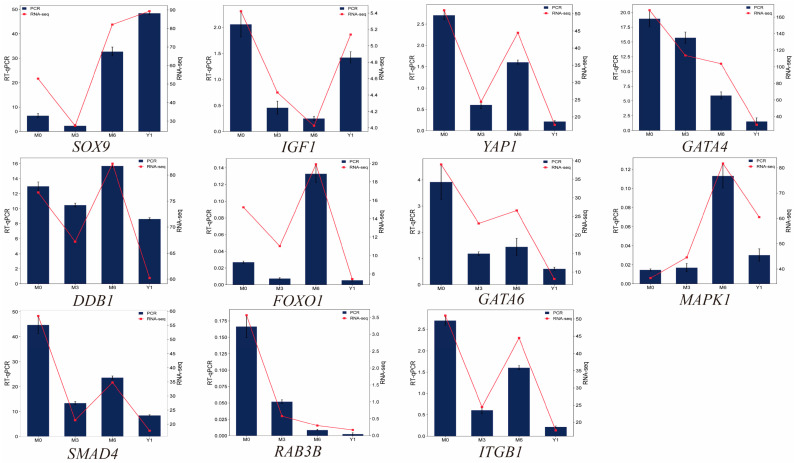
Validation of differentially expressed genes by RT-qPCR. The bar charts represent RT-qPCR and the line charts represent RNA-seq.

## Data Availability

Data are available upon request due to privacy/ethical restrictions.

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
