# Peer review of "Transcriptomic Study of Different Stages of Development in the Testis of Sheep"

_animals, 2024, doi:10.3390/ani14192767_

Round 1

Reviewer 1 Report

Comments and Suggestions for Authors

I was invited to review the 3167687 manuscript entitled "Transcriptomic Study of Different Stages of Development in the Testis of Sheep" for publication in Animals (Animal Reproduction).

The work aimed to explore candidate genes and pivotal pathways influencing fecundity in F1 hybrid of Southdown x Hu sheep testes across four developmental milestones.

Here are some questions that should be addressed in future submissions:

Abstract:

It would be interesting to include the results of the “Histological Observation of Testis” briefly.

Introduction:

Lines 42-92: this first part of the introduction could be shortened. It contains a lot of information that could be consulted in the references if you want to go deeper into the characteristics of the breeds used in this experiment, for example.

Line 65: the authors should use a ovine species reference.

Line 66: should be “understand” not “under-stand”.

Line 75: should be “renowned breed” not “renowned breed renowned”.

Line 95: should be “expression” not “ex-pression”.

Line 108: should be “provide” not “pro-vide”.

Line 105: End the introduction with the objective of the work. This study aims to…

Line 107: “Transcriptome analysis of testicular tissue from Southdown × Hu F1 sheep at 0, 3, 6, and 12 months of age using RNA-Seq and bioinformatics techniques, and these findings pro-vide new insights into the regulatory mechanisms of testicular development and spermatogenesis in sheep” should be rewritten.

Regarding material and methods:

Line 116: to use an n = 3, has any computer software been used to establish that these are minimum animals per experimental group? If so, the authors should indicate this.

Line 128: should be “Qinghuan” not “Qing huan”.

Line 134: the authors should remove commas.

In the results section:

Line 193: The authors should write in the manuscript the numerical data of the most relevant and significant comparisons.

Line 215: Figure B. Arrange the columns and the legend in the same way. The figure should be larger as well as its legend. As a recommendation, I would add a Y-axis to the right of the graph to not divide the Y-axis into three parts.

Line 272: the authors should divide this sentence: “There were significant differences in the expression of up-regulated-(red) and down-(blue) regulatory genes black indicates no differential expression” in two to improve its comprehension.

In the discussion section:

Line 500: I don’t understand these words: “Guo-min zhang etc.”

I would like to find in the discussion a paragraph in which the practical approach of this study was sought. In what practical situations these findings would be of use in improving fertility and, thus, herd profitability.

Conclusions:

The conclusions must respond to the stated objective… The first part of the conclusion is similar to the results section. The results should not be repeated in the conclusion.

Supplementary Materials: I can’t find the supplementary material to review.

Author Response

Dear Editor and Reviewers,

Thank you very much for taking the time to review our manuscript entitled “Transcriptomic Study of Different Stages of Development in the Testis of Sheep”. Your comments and suggestions have been invaluable in improving the quality of our work. We have carefully considered each of your points and have made revisions accordingly.

Abstract:

1.Question: It would be interesting to include the results of the “Histological Observation of Testis” briefly.

Respond: As the reviewer suggests, we have included the content of testicular histological observations(L28-32).

Introduction:

  1. Question: the authors should use a ovine species reference.

Respond: Ovine species references have been cited (L52, L54).

  1. Question: should be “understand” not “under-stand”.

Line 75: should be “renowned breed” not “renowned breed renowned”.

Line 95: should be “expression” not “ex-pression”.

Line 108: should be “provide” not “pro-vide”.

Respond: All have been modified as required.

  1. Question: Line 105: End the introduction with the objective of the work. This study aims to…

“Transcriptome analysis of testicular tissue from Southdown × Hu F1 sheep at 0, 3, 6, and 12 months of age using RNA-Seq and bioinformatics techniques, and these findings pro-vide new insights into the regulatory mechanisms of testicular development and spermatogenesis in sheep” should be rewritten.

Respond: Modify it as per the instructions to “We performed a transcriptome analysis of testicular tissue from Southdown × Hu F1 sheep at various developmental stages (0, 3, 6, and 12 months of age) utilizing RNA-Seq and bioinformatics methodologies. The results of this analysis offer fresh insights into the regulatory mechanisms that govern testicular development and spermatogenesis in sheep” (L78- L82).

Regarding material and methods:

  1. Question: use an n = 3, has any computer software been used to establish that these are minimum animals per experimental group? If so, the authors should indicate this.

Respond: Three sheep with the smallest differences in body weight and phenotypic characteristics during the same developmental period were selected for castration.

  1. Question: should be “Qinghuan” not “Qing huan”.

the authors should remove commas.

Respond: We made the necessary changes to these two sections as requested. (L101, L107).

In the results section: 

  1. Question: The authors should write in the manuscript the numerical data of the most relevant and significant comparisons.

Respond: We simplified and reorganized the results to make necessary adjustments. (L166, L173).

  1. Question: Figure B. Arrange the columns and the legend in the same way. The figure should be larger as well as its legend. As a recommendation, I would add a Y-axis to the right of the graph to not divide the Y-axis into three parts.

Respond: We carefully considered your comments, but found that adding the Y-axis on the right side did not stand out, so we divided it into 7 bar graphs. For details, see Figure 1B.

  1. Question: the authors should divide this sentence: “There were significant differences in the expression of up-regulated-(red) and down-(blue) regulatory genes black indicates no differential expression” in two to improve its comprehension.

Respond: We modified as required“Volcano map of differentially expressed genes (DEGs).  The red, blue, and black dots in the figure represent significantly up-regulated, down-regulated, and unchanged transcripts, respectively.”

In the discussion section:

10.Question: I don’t understand these words: “Guo-min zhang etc.”

I would like to find in the discussion a paragraph in which the practical approach of this study was sought. In what practical situations these findings would be of use in improving fertility and, thus, herd profitability.

Respond: Thank you very much for your suggestions. There is an error here, and we have deleted it.

11.Question: The conclusions must respond to the stated objective… The first part of the conclusion is similar to the results section. The results should not be repeated in the conclusion.

Respond: We have optimized and streamlined the conclusion section according to your suggestions, see details L483- L494.

  1. Question: Supplementary Materials: I can’t find the supplementary material to review.

Respond: Thank you for reminding us. Our supplementary material is a word document, and we will upload it again.

In summary, we have made substantial revisions to the manuscript based on your feedback. We believe that these changes have improved the overall quality and clarity of our work. We appreciate your thoughtful comments and look forward to hearing from you again. 

Thank you once again for your time and expertise in reviewing our manuscript. 

Sincerely, 

Binpeng Xi

Tel: +86-13659401400 Email: w5562080w@163.com

Reviewer 2 Report

Comments and Suggestions for Authors

The presented work is performed at the highest level and is of high relevance and practical importance for practical sheep breeding and scientific community. The review did not reveal any critical comments, but there are some recommendations regarding the introduction to the paper. In the introduction it is desirable to reduce a fragment of the text describing the stages of formation of reproductive organs in rams in ontogenesis (lines 45-67), because it contains well-known provisions. It is also recommended to shorten the material describing the Southdown and Hu breeds (lines 68-92). In the final paragraph of the introduction (lines 93-110), it is recommended to cite sources on full transcriptome studies of reproductive organ samples from productive farm animals, in addition to information on these studies on mice and Drosophila. 

Author Response

Dear Editor and Reviewers,

Thank you very much for taking the time to review our manuscript entitled “Transcriptomic Study of Different Stages of Development in the Testis of Sheep”. Your comments and suggestions have been invaluable in improving the quality of our work. We have carefully considered each of your points and have made revisions accordingly.

1.Question: In the introduction it is desirable to reduce a fragment of the text describing the stages of formation of reproductive organs in rams in ontogenesis (lines 45-67), because it contains well-known provisions.

Respond: Based on your valuable comments, we have streamlined this section and removed some redundant and unnecessary sentences. For details, see (lines 44-51).

2.Question: It is also recommended to shorten the material describing the Southdown and Hu breeds (lines 68-92).

Respond: We have streamlined the materials of the two varieties according to your requirements and introduced the literature related to germplasm resources of this variety for reference. see (lines 52-55).

3.Question: In the final paragraph of the introduction (lines 93-110), it is recommended to cite sources on full transcriptome studies of reproductive organ samples from productive farm animals, in addition to information on these studies on mice and Drosophila. 

Respond: After taking your comments into consideration, we introduced the complete transcriptome study of Tibetan sheep, bovine and yak reproductive organ samples (lines 69).

In summary, we have made substantial revisions to the manuscript based on your feedback. We believe that these changes have improved the overall quality and clarity of our work. We appreciate your thoughtful comments and look forward to hearing from you again. 

Thank you once again for your time and expertise in reviewing our manuscript. 

Sincerely, 

Binpeng Xi

Tel: +86-13659401400 Email: w5562080w@163.com

Reviewer 3 Report

Comments and Suggestions for Authors

The paper titled Transcriptomic Study of Different Stages of Development in the Testis of Sheep appears to be a well-structured scientific study. It includes comprehensive sections such as the introduction, materials and methods, results, and discussion, with detailed transcriptomic analysis across different developmental stages of sheep testes. The study employs RNA-Seq, RT-qPCR, and bioinformatics techniques to investigate gene expression patterns, differential gene expression, and key pathways influencing spermatogenesis.

 Here some minor report that could imporve overall quality, afterthat the paper could be, in my opinion, accepted as is.

Line 155: “OAR3.1” Only a details that have to taken in count or explained: Several newer versions of the sheep genome assembly, such as Oar_v4.0 and Oar_rambouillet_v1.0, are available, offering improved accuracy and completeness. Using an older genome assembly like OAR3.1 could limit the study's relevance, especially when more recent versions provide better annotations, which could enhance the resolution of gene identification and functional analysis.

It would be beneficial for the authors to justify their choice of OAR3.1. They should explain whether there were specific reasons for using this assembly (e.g., compatibility with certain tools or previous studies) or if it was simply an oversight. Suggesting the authors to reanalyze their data using a more recent assembly could be a valid recommendation, as it might improve the reliability and accuracy of their findings.

Line 83-92: The authors do mention in the introduction why they used F1 hybrid animals (Southdown × Hu sheep). Specifically, they highlight that hybridization is a key method for generating heterosis, which can enhance traits compared to the parent strains. In this case, the F1 hybrid shows superior traits in body size, growth rate, daily gain, and meat quality, as well as improved reproductive performance when compared to pure Hu sheep.

However, the explanation could be strengthened. While the authors justify using F1 hybrids based on general traits like growth and meat quality, they could provide more specific reasons as to why these hybrids were chosen for a transcriptomic study on testicular development. For instance, they could elaborate on how these F1 hybrids might offer unique insights into gene expression during spermatogenesis or whether the hybrid's heterosis affects reproductive traits in a way that makes them particularly useful for this kind of research.

In summary, while the paper touches on why F1 hybrids were used, a more detailed explanation related to the specific goals of studying testicular development and spermatogenesis would make the reasoning more convincing.

Some spare comments:

Here are a few points that might warrant further consideration or clarification in the review process:

  1. Statistical Robustness: The study reports multiple significant findings, but a deeper look at the statistical methods (especially in terms of the p-values and fold changes) could be beneficial. While they used the Benjamini-Hochberg procedure for multiple testing corrections, ensuring the robustness of statistical significance (especially for small sample sizes) could be discussed more explicitly.
  2. Visual Clarity of Figures: Some figures, such as the gene expression trend charts and protein-protein interaction networks, are crucial for understanding the study's conclusions. Ensuring that these figures are clear and easy to interpret (especially for readers unfamiliar with bioinformatics) could enhance the overall readability.
  3. GO and KEGG Pathway Analyses: While the enrichment analyses are well-done, a more detailed exploration of how these pathways interact or overlap could strengthen the argument. For instance, the relationship between MAPK signaling, PI3K-Akt signaling, and reproductive development might benefit from further elaboration.

Author Response

Dear Editor and Reviewers,

Thank you very much for taking the time to review our manuscript entitled “Transcriptomic Study of Different Stages of Development in the Testis of Sheep”. Your comments and suggestions have been invaluable in improving the quality of our work. We have carefully considered each of your points and have made revisions accordingly.

1.Question: “OAR3.1” Only a details that have to taken in count or explained: Several newer versions of the sheep genome assembly, such as Oar_v4.0 and Oar_rambouillet_v1.0, are available, offering improved accuracy and completeness. Using an older genome assembly like OAR3.1 could limit the study's relevance, especially when more recent versions provide better annotations, which could enhance the resolution of gene identification and functional analysis. It would be beneficial for the authors to justify their choice of OAR3.1. They should explain whether there were specific reasons for using this assembly (e.g., compatibility with certain tools or previous studies) or if it was simply an oversight. Suggesting the authors to reanalyze their data using a more recent assembly could be a valid recommendation, as it might improve the reliability and accuracy of their findings.

Respond: Thank you very much for your valuable comments. We have corrected the error here in the article. The reference genome used in this study is "Qar rambouillet v1.0".

2.Question: The authors do mention in the introduction why they used F1 hybrid animals (Southdown × Hu sheep). Specifically, they highlight that hybridization is a key method for generating heterosis, which can enhance traits compared to the parent strains. In this case, the F1 hybrid shows superior traits in body size, growth rate, daily gain, and meat quality, as well as improved reproductive performance when compared to pure Hu sheep. However, the explanation could be strengthened. While the authors justify using F1 hybrids based on general traits like growth and meat quality, they could provide more specific reasons as to why these hybrids were chosen for a transcriptomic study on testicular development. For instance, they could elaborate on how these F1 hybrids might offer unique insights into gene expression during spermatogenesis or whether the hybrid's heterosis affects reproductive traits in a way that makes them particularly useful for this kind of research.In summary, while the paper touches on why F1 hybrids were used, a more detailed explanation related to the specific goals of studying testicular development and spermatogenesis would make the reasoning more convincing.

Respond: Thank you very much for your valuable advice. Hu sheep is famous for its excellent fecundity and multi-parity, and Southdown sheep has excellent meat quality. Through the hybridization of the two sheep, their respective advantages are concentrated on the offspring generation, but the breeding data of the offspring generation of the hybrid are very few, and we can only obtain relevant published data on the meat quality. However, our study provides the foundation for the first transcriptomic analysis of reproductive performance in the offspring generation of the two hybrids.

3.Question: The study reports multiple significant findings, but a deeper look at the statistical methods (especially in terms of the p-values and fold changes) could be beneficial. While they used the Benjamini-Hochberg procedure for multiple testing corrections, ensuring the robustness of statistical significance (especially for small sample sizes) could be discussed more explicitly. 

Respond: Thank you for your valuable comments, we have re-optimized the statistical method.

4.Question: Some figures, such as the gene expression trend charts and protein-protein interaction networks, are crucial for understanding the study's conclusions. Ensuring that these figures are clear and easy to interpret (especially for readers unfamiliar with bioinformatics) could enhance the overall readability. 

Respond: We have processed the pictures according to your comments and improved their clarity.

3.Question: While the enrichment analyses are well-done, a more detailed exploration of how these pathways interact or overlap could strengthen the argument. For instance, the relationship between MAPK signaling, PI3K-Akt signaling, and reproductive development might benefit from further elaboration. 

Respond: Overlap analysis of enriched pathways was performed to elucidate their roles in testicular development and spermatogenesis. (lines 398-453).

In summary, we have made substantial revisions to the manuscript based on your feedback. We believe that these changes have improved the overall quality and clarity of our work. We appreciate your thoughtful comments and look forward to hearing from you again. 

Thank you once again for your time and expertise in reviewing our manuscript. 

Sincerely, 

Reviewer 4 Report

Comments and Suggestions for Authors

The manuscript, titled "Transcriptomic Study of Different Stages of Development in the Testis of Sheep," offers a thorough examination of testicular development and spermatogenesis using RNA-Seq, a technique for analyzing gene expression. The methodology employed to analyze gene expression across four discrete developmental stages is skillfully conducted and provides vital insights into the regulatory processes that govern male reproduction. The praiseworthy aspects include the discovery of genes that are expressed differently and the enrichment of important pathways such as cAMP, MAPK, ECM-receptor interaction, PI3K-Akt, and FOXO signaling. The findings successfully emphasize the intricate molecular alterations linked to testis maturation and provide a strong basis for comprehending the shift from immature to mature testes. The identification of heightened alternative splicing occurrences at the later stages (M6 and Y1) introduces a significant aspect to your investigation and emphasizes the ever-changing nature of gene control during testicular development. The emphasis on crucial genes such as GATA4, SOX9, and MAPK1 is notably important and offers a valuable structure for forthcoming studies aimed at enhancing reproductive methods. In general, the work is well-organized and makes a significant contribution to the subject. I have appended some remarks and recommendations in the accompanying document to assist in further refining and fortifying your article. 

Author Response

Dear Editor and Reviewers,

Thank you very much for taking the time to review our manuscript entitled “Transcriptomic Study of Different Stages of Development in the Testis of Sheep”. Your comments and suggestions have been invaluable in improving the quality of our work. We have carefully considered each of your points and have made revisions accordingly.

  1. The abstract tries to include too many specifics, such as the exact number of DEGs identified in each comparison, which may overwhelm readers. Abstracts should be concise and focus on the most critical information, leaving detailed data for the main text.

Respond: Thank you for your valuable comments, we have simplified the original abstract, see the details L24-L39.

2.Question: The type, dose, and administration method of the anesthesia used before castration are not specified. This information is essential to ensure that the procedure was humane and to allow for replication.

Respond: Here's how to deal with it “Anesthesia was induced via intramuscular diazepam (410 mg) and scopolamine (90.3 mg), subsequently followed by intravenous thiopental sodium (A10-20 mg/kg)” L97-L99.

3.Question: While the use of RNA/DNA sample protector and storage conditions are mentioned, the handling time between testis removal and freezing in liquid nitrogen is not specified. RNA and protein integrity can be highly time-sensitive, so this information is crucial.

Respond: Thank you for your valuable comments. The relevant treatment methods are as follows “Dissect each sheep's testis longitudinally, collect right testicular tissue within 3 mins, freeze immediately in liquid nitrogen for 15 mins, store at -80°C, then extract total RNA and protein.” L90-L92.

4.Question: The fixation in 2.5% glutaraldehyde is mentioned, but details on the duration of fixation, subsequent processing steps for paraffin embedding, and any specific staining techniques to be used are missing. 

Respond: We supplemented the methods and procedures of the relevant sections, and the subsequent staining techniques are detailed in 2.3. L99-L101.

5.Question: The use of Image Pro-Plus 6.0 software is appropriate, but the description lacks information on how calibration was performed to ensure accurate measurements. Also, the criteria for selecting "randomly selected seminiferous tubules" should be clarified to avoid potential bias.

Respond: We have revised it according to your comments, see details L105-112.

6.Question: Some sentences are overly complex and could be simplified for better readability.

Respond: We have simplified and optimized the results presented here, as described in detail L167-174.

7.Question: The description of (distinct cavities) within the seminiferous tubules at 1 year of age could be elaborated to explain whether these cavities are pathological or part of normal development.

Respond: Thank you for your comments. In early development, seminiferous tubules have almost no lumen and the seminiferous tubule epithelium is composed of spermatogonia and Sertoli cells. From puberty, seminiferous tubules develop rapidly and lumens begin to appear and rapidly enlarge. It tends to stabilize in adulthood. In our selection of materials, samples taken from healthy sheep were chosen, and cavities within seminiferous tubules are part of normal development.

8.Question: How do the changes in the number and function of Sertoli cells between the different developmental stages influence spermatogenesis and overall testicular function?

Respond: Regarding the question you raised, we have made a supplementary explanation in the discussion section “Sertoli cells provide energy and nutrients for spermatogenesis and secrete a variety of substances that participate in the differentiation and maturation of spermatogenic cells to ensure the occurrence of sperm. The number of Sertoli cells stabilizes in puberty. Reduction in fetal Sertoli cells significantly decreases testicular cord number, likely due to loss of basement membrane components they secrete. In mice, Sertoli cells proliferate post-puberty arrest, achieving balance. Birth proliferation changes irreversibly alter this number, supporting the need for adequate Sertoli cells for testicular development.” L328-334.

9.Question: How do changes in the diameter, area, and epithelial thickness of seminiferous tubules correlate with the stages of spermatogenesis in sheep?

Respond: Regarding the question you raised, we have made a supplementary explanation in the discussion section “Recent studies have shown that the migration of spermatogenic cells in the seminiferous epithelium is associated with the remodeling of anchor junctions between cells during spermatogenesis, which are involved in spermatogenesis and sperm release. Study shows higher germ cell count & seminiferous tubule diameter, area and epithelial thickness in M6 and Y1 stages vs. others. We hypothesize a positive correlation among these factors, reflecting internal structural development and germ cell proliferation, these changes provide sufficient space and nutritional support for germ cells, thereby promoting spermatogenesis.” L343-350.

10.Question: The hypothesis presented at the end of the section could be more clearly articulated. The statement (the reproductive system developed full spermatogenic ability and related germ cells increased greatly after somatic maturation) is somewhat ambiguous. It would be better to specify the exact developmental changes hypothesized to occur.

Respond: We have rewritten this part according to the suggestions you gave, see details L341-351.

11.Question: The section mentions the importance of DDB1 but does not delve into the specific molecular pathways or interactions through which DDB1 influences Sertoli cell proliferation and testicular development. A more detailed exploration of these pathways would provide greater clarity.

Respond: We have reformulated the relevant pathways by which DDB1 functions in germ cells, as detailed in the Discussion section, DDB1 is ubiquitally expressed in almost all testicular cell types, and previous studies have shown that it is essential for germ cell development. DDB1 regulates the ubiquitination and degradation of cell cycle inhibitors such as p27 and affects the progression of cell cycle, thereby ensuring the smooth progress of spermatogenesis. We hypothesized that DDB1 plays a key role in testicular development in sheep, specifically, a role that is essential for both testicular development and spermatogenesis. L374-381.

12.Question: The conclusion mentions several signaling pathways (cAMP, PI3K-Akt, FOXO, HIF-1) without providing specific insights into how these pathways contribute to the observed gene expression changes or their roles in spermatogenesis. This generalization diminishes the impact of the findings.

Respond: Overlap analysis of enriched pathways was performed to elucidate their roles in testicular development and spermatogenesis. L398-453.

In summary, we have made substantial revisions to the manuscript based on your feedback. We believe that these changes have improved the overall quality and clarity of our work. We appreciate your thoughtful comments and look forward to hearing from you again. 

Thank you once again for your time and expertise in reviewing our manuscript. 

Sincerely, 

Binpeng Xi

Tel: +86-13659401400 Email: w5562080w@163.com